# African bushpigs exhibit porous species boundaries and appeared in Madagascar concurrently with human arrival

Several African mammals exhibit a phylogeographic pattern where closely related taxa are split between West/Central and East/Southern Africa, but their evolutionary relationships and histories remain controversial. Bushpigs (*Potamochoerus larvatus*) and red river hogs (*P. porcus*) are recognised as separate species due to morphological distinctions, a perceived lack of interbreeding at contact, and putatively old divergence times, but historically, they were considered conspecific. Moreover, the presence of Malagasy bushpigs as the sole large terrestrial mammal shared with the African mainland raises intriguing questions about its origin and arrival in Madagascar. Analyses of 67 whole genomes revealed a genetic continuum between the two species, with putative signatures of historical gene flow, variable $F_{ST}$ values, and a recent divergence time (<500,000 years). Thus, our study challenges key arguments for splitting *Potamochoerus* into two species and suggests their speciation might be incomplete. Our findings also indicate that Malagasy bushpigs diverged from southern African populations and underwent a limited bottleneck 1000-5000 years ago, concurrent with human arrival in Madagascar. These results shed light on the evolutionary history of an iconic and widespread African mammal and provide insight into the longstanding biogeographic puzzle surrounding the bushpig's presence in Madagascar.

The African Suidae lineage contains six recognised extant species: common warthog (*Phacochoerus africanus*), desert warthog (*Ph. aethiopicus*), giant forest hog (*Hylochoerus meinertzhageni*), wild boar (*Sus scrofa*), red river hog (*Potamochoerus porcus*), and bushpig (*P. larvatus*)[1,2]. There are several unresolved aspects of the evolutionary history of African pigs, including a controversial timeline for their divergence which stems from molecular estimates that predate fossil records by millions of years and the unresolved role of gene flow between lineages[3,4]. The two members of the genus *Potamochoerus* – red river hog and bushpig – were historically considered conspecific, despite considerable morphological differences[5,6]. They occur parapatrically in West/Central (W/C) Africa and East/Southern (E/S) Africa with some populations possibly having abutting or slightly overlapping ranges[2] (Fig. 1a). Based primarily on morphological differences

and a lack of evidence that these taxa hybridise at contact, Grubb proposed the currently accepted nomenclature, regarding them as two distinct species[7,8].

The distribution of the two *Potamochoerus* species is similar to that found in several other African mammals that have ecologically comparable sister (sub)species pairs. The W/C and E/S divide has been highlighted as one of the most important biogeographic patterns in Africa and is potentially connected to the initial divergence between hominins and apes[9], even if at a different time scale. This evolutionary divergence into W/C and E/S lineages occurred relatively recently for some mammalian taxa such as the African buffalo (*Syncerus caffer*)[10] and the lion (*Panthera leo*)[11] leading to subspeciation, whereas in other taxa an older split led to full speciation, e.g., in African elephants (*Loxodonta* spp.)[12] and baboons (*Papio* spp.)[13]. For all species

✉e-mail: ida@bio.ku.dk; aalbrechtsen@bio.ku.dk; rheller@bio.ku.dk

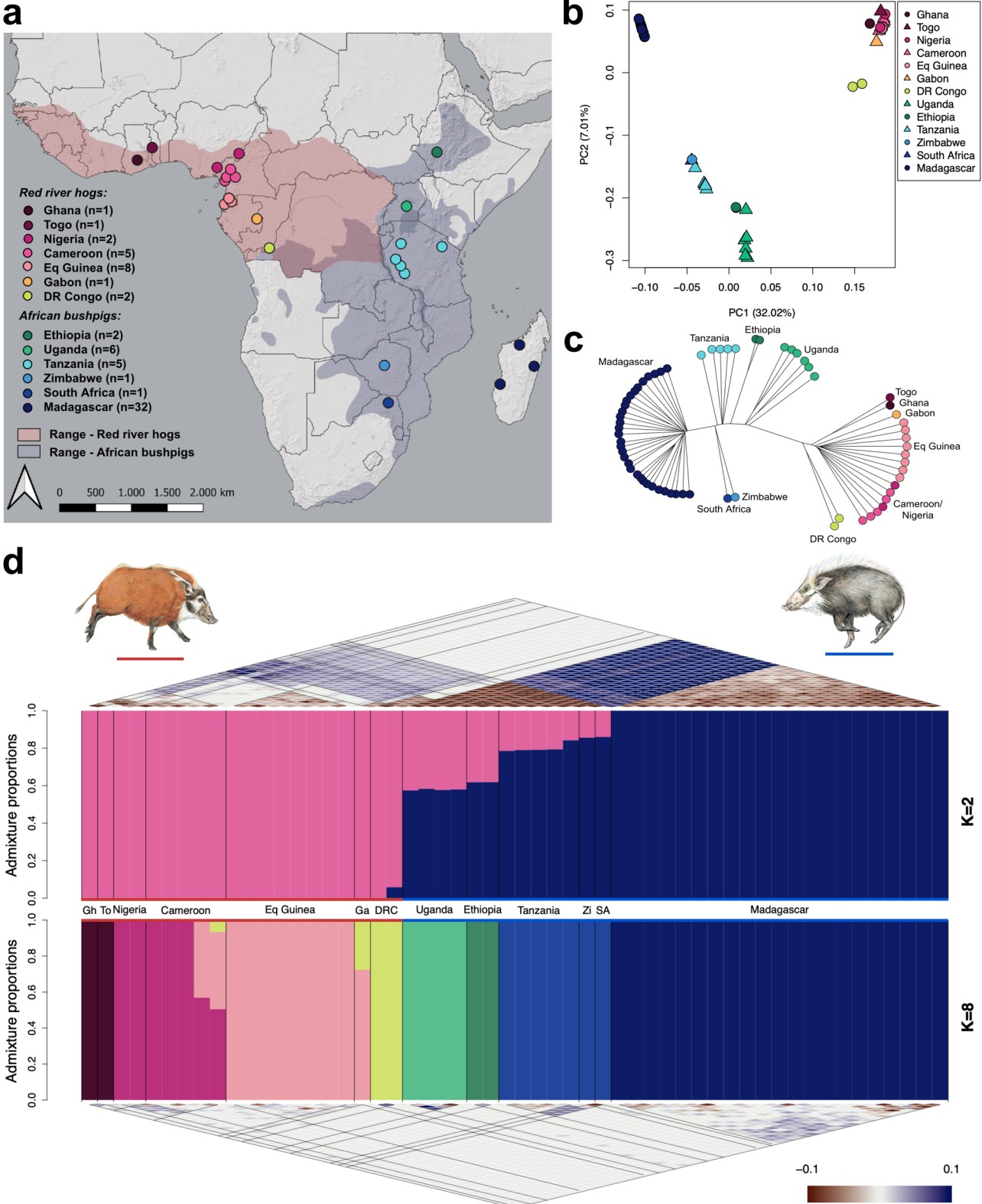

**Fig. 1 | Sampling and population structure of red river hogs and bushpigs.**
**a** Sampling map of all 67 pig individuals used within this study, coloured by country of origin. Ranges for red river hogs and bushpigs are shaded in red and blue, respectively[52,104]. **b** Principal component analysis (PCA) for 67 pigs, showing the first two principal components, coloured by country. **c** Unrooted neighbour-joining tree based on pairwise identity-by-state (IBS), coloured by country (*n* = 67). **d** Inferred ancestry proportions for 54 unrelated samples using NGSadmix[32], assuming *K* = 2 (upper barplot) and *K* = 8 (bottom barplot). Coloured bars indicate estimated admixture proportions with the number of ancestries equal to *K*. Coloured lines above and below population labels and illustrations indicate species designations; red–red river hogs, blue–bushpigs. Pairwise correlations of residuals as estimated by evalAdmix[33] are shown above and below the respective NGSadmix barplots ranging from −0.1 to 0.1 (colour scale). Gh Ghana, To Togo, Ga Gabon, DRC Democratic Republic of Congo, Zi Zimbabwe, SA South Africa. Bushpig and red river hog illustrations are used with express permission from the author, Jonathan Kingdon[2].

mentioned above, a hybrid zone has been identified where the ranges of diverged lineages overlap. Although possible hybridisation between the two *Potamochoerus* species has been suggested[7], the evolutionary connection and the geographic context of a likely suture zone are still poorly understood[5,14]. Recent range contractions limit the overlap of the two species ranges to Uganda and the Democratic Republic of Congo (DR Congo). However, South Sudan and possibly Ethiopia were part of a suture zone in the recent past, when the red river hog range extended further towards the east (Fig. 1a)[5]. The evolutionary processes occurring in these suture zones, found recurrently across many taxa, e.g., in western Uganda[10,15], are of particular interest for understanding speciation and the phylogeography of African mammals in general.

Bushpig populations in Madagascar provide an interesting case of possible human-mediated range expansion. The bushpig represents a biogeographic anomaly in being the only large, wild terrestrial mammal to be shared between the African mainland and the island of Madagascar[16]. These land masses separated around 150 million years ago, leading to largely divergent flora and fauna[17,18]. For some Malagasy taxa, such as lemurs, it has long been debated whether colonisation of Madagascar could have taken place through island hopping or temporal land bridges[19]. It is now commonly accepted that some of these taxa arrived on Madagascar by rafting on floats of vegetation and that successful colonisation events and subsequent radiation led to the diversity seen today[20,21]. For bushpigs, it has been proposed that the most plausible explanation is that they were introduced to Madagascar by humans, possibly through the Comoros Islands[22,23]; however, this has not been conclusively verified. Humans are believed to have been present in Madagascar no earlier than 11,000 years ago[24], with some authors claiming that there is no proof of human presence older than 2000 years[25]. Nevertheless, most authors agree that there were no significant numbers of humans until 1000–1500 years ago with the arrival of populations from Southeast Africa (Bantu speakers) and Southeast Asia (Austronesian speakers)[24,26,27]. Radiocarbon dating of archaeological remains suggests that bushpigs, as well as zebu, sheep, and goats were established in Southwest Madagascar between 700–1200 years ago; this estimate, however, may have been biased by the scarce data available for Malagasy bushpigs[28]. To our knowledge, there is only one study which attempted to estimate the arrival of bushpigs on Madagascar based on genetic data – this study suggested a split 480 thousand years ago (kya) based on mitochondrial DNA (mtDNA) divergence times, which is not in line with a proposed human-mediated introduction to the island[29]. In addition to their time of arrival, the source population for Malagasy bushpigs is still debated, where despite detailed morphological studies, these have been unable to conclusively resolve their mainland origin[8,30]. Existing genetic data tentatively suggest an origin from Central Southern Africa[29]. If bushpigs were indeed introduced to Madagascar by humans, it presents another suite of questions as there is no archaeological or other evidence of bushpig domestication ever occurring despite them being an important protein source for many rural communities[31]. For example, the transportation of such a large non-domesticated mammal over the wide (>400 km) Mozambique channel remains an unsolved mystery and may provide an indirect indication that populations located on the southeastern African coast mastered oceanic travel beyond fishing[29]. Alternatively, a much older divergence time could provide indirect proof of a very early African presence in Madagascar.

In this study, we present population genomic analyses of 67 whole genomes from *Potamochoerus*, including 32 bushpigs from Madagascar. We investigate their population structure and genetic diversity and infer gene flow between the two taxa. We also estimate the degree of evolutionary divergence between the bushpig and red river hog relative to co-occurring species that represent incomplete or full speciation. Finally, we address the question of when and from where in Africa the bushpig colonised Madagascar, clarifying several details regarding the origin of Malagasy bushpigs. Our analyses present insights that improve our understanding of African biogeography and help settle a major question regarding prehistoric human activities shaping biodiversity patterns in Africa.

## Results

### Sampling and filtering

Whole genome sequencing data were generated for 71 *Potamochoerus* samples across the two species' ranges, including 23 red river hogs and 48 bushpigs (range 3×–101×, mean ≈12.8×; Fig. 1a; Supplementary Data 1). All samples were mapped to a chromosome-level common warthog reference genome and site filtering applied to reduce downstream errors (see Methods; Supplementary Data 2). Two red river hog samples, from Cameroon and DR Congo, were excluded due to high sequencing error rates (Supplementary Fig. 1). Four samples, two from Equatorial Guinea (Eq Guinea) and two from Ethiopia, were deemed to originate from the same individual and were merged into one sample for each respective locality (Supplementary Fig. 2). A total of 13 samples were first degree relatives (parent-offspring or full siblings), of which 11 were from Madagascar and two were from Uganda (Supplementary Data 1). Depending on the specific requirements of various downstream analyses, these samples were sometimes excluded. In summary, whole genome sequencing data from 67 pigs from 13 countries were analysed, of which 54 were not closely related, including 18 that were sequenced at medium-high depth (≥14×; Fig. 1a; Supplementary Data 1). A summary of datasets, analyses, and methods used in this study is described in Supplementary Data 3.

### Localised population structure and no recent admixture between red river hogs and bushpigs

We first aimed to gain insights into the population structure of red river hogs and bushpigs, specifically examining genetic differentiation between populations of both species[32]. Principal component analysis (PCA) revealed that the first two principal components exhibited a spatial distribution pattern reflecting the taxonomic and geographic origins of the sampled pigs (Fig. 1b). All the red river hog samples clustered together, yet the Congolese individuals were closer to the bushpigs than the other red river hogs. Malagasy samples formed a separate cluster from the other bushpigs. A neighbour-joining tree using identity-by-state delineated a clear division between red river hogs and bushpigs, displaying a basal split between the two groups (Fig. 1c). The tree also revealed more localised population substructure, including the Malagasy samples forming a clade separate from the other bushpigs.

We next inferred ancestry proportions within both species to further explore population substructure. Assuming the number of ancestral populations was 2 ($K = 2$), the result largely aligned with the pattern observed in PC1-PC2 (Fig. 1d). Notably, we did not observe a clear separation of red river hogs and bushpigs at $K = 2$, even when excluding Madagascar samples (Supplementary Fig. 3). It is worth noting that evalAdmix[33] indicated unresolved substructure, suggesting that this pattern should not be literally interpreted as the result of admixture and these numbers as admixture proportions. We obtained a much better fit by assuming a higher number of ancestral populations ($K = 8$, Fig. 1d; Supplementary Fig. 4) and were able to assign most geographic locations to their own ancestral population. However, this analysis did not reveal evidence of recent gene flow between bushpigs and red river hogs.

### Moderate differentiation and gene flow between red river hogs and bushpigs through Uganda

Genetic differentiation between all pairs of populations was assessed using Hudson's $F_{ST}$ and $D_{xy}$ between unrelated individuals (Fig. 2a)[34,35]. $F_{ST}$ values generally correlated with geographic distance (Fig. 2a; Supplementary Fig. 5). Notably, Ethiopia and Madagascar exhibited

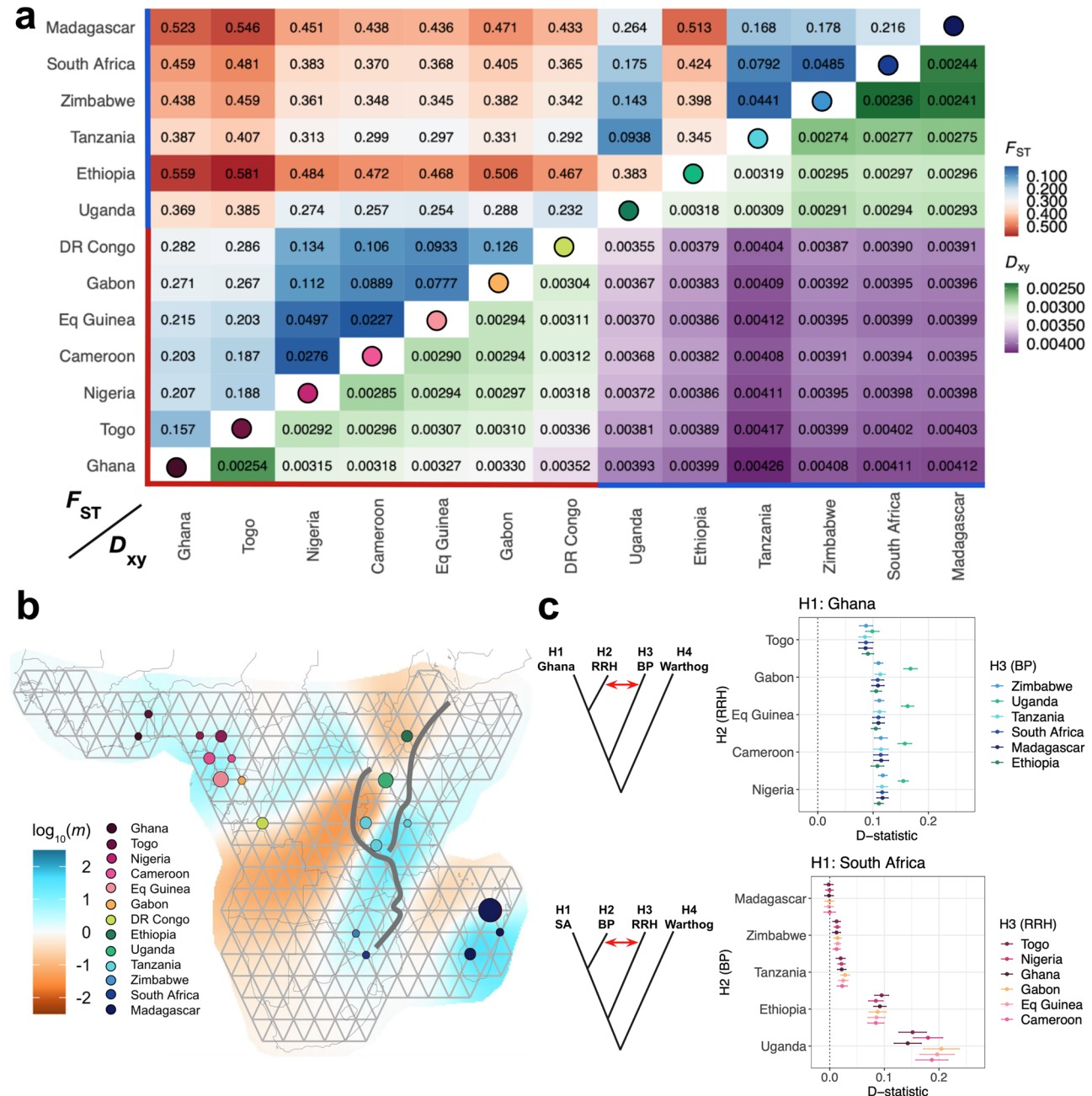

**Fig. 2 | Genetic differentiation and gene flow between bushpigs and red river hogs. a** Genetic differentiation as described by pairwise Hudson's $F_{ST}$[34] (red–blue) and $D_{xy}$[35] (green–purple) for 54 unrelated individuals, rounded to three significant figures. Circles on the diagonal correspond to populations as in (**b**). Coloured lines above and next to population labels indicate species designations; red–red river hogs, blue–bushpigs. **b** Estimated effective migration surfaces using EEMS[36]. Circles are coloured by country of origin. $\log_{10}(m)$ describes the effective migration rate relative to the overall migration rate across indicated regions. The East African Rift Valley is depicted by grey lines. **c** D-statistics using medium-high depth (≥14×) samples (n = 18) with the common warthog as an outgroup, constructed as D (H1, H2, H3, Warthog). A significant non-zero positive value, as depicted by the red arrow in the graphic for each panel, provides evidence for gene flow between H3 and H2, relative to H1 (i.e., H2 is closer to H3 than H1)[105]. Upper panel – D-statistics testing for gene flow signals between bushpigs (H3) and non-Ghana red river hogs (H2). Lower panel – D-statistics testing for gene flow signals between red river hogs (H3) and non-South-African bushpigs (H2). Data are presented as the estimated D-statistic ± three standard errors. RRH red river hog, BP bushpig, SA South Africa.

higher $F_{ST}$ values (0.345–0.581 and 0.168–0.546, respectively). Excluding these populations, $F_{ST}$ values within red river hog populations ranged from 0.023–0.286, while $F_{ST}$ values within bushpigs were between 0.044 and 0.175. When comparing across species, $F_{ST}$ excluding Ethiopia was higher (0.232–0.546), though not markedly so when compared to the most differentiated population pairs in within-species comparisons. $D_{xy}$ values exhibited a similar trend, displaying increased nucleotide diversity between species relative to within-species comparisons (Fig. 2a). As with $F_{ST}$, $D_{xy}$ also correlated with geographic distance with the exception of Tanzania; this population had increased $D_{xy}$ relative to other populations. Notably, $D_{xy}$ for Ethiopia was similar to those between other bushpig and red river hog populations, suggesting that the high $F_{ST}$ observed for Ethiopia was likely driven by lower within-population diversity. When comparing across species, $D_{xy}$ was lowest for populations that were geographically central, with the Ugandan population exhibiting the lowest

between-species $D_{xy}$ for all bushpigs and the Congolese the lowest $D_{xy}$ for all red river hogs. Additionally, the lowest $D_{xy}$ between species was observed between the Ugandan and Congolese populations (0.00355), similar to within-species $D_{xy}$ comparisons for Ghana and DR Congo (0.00352).

Given the observed $F_{ST}$ and $D_{xy}$ values, we explored spatial population structure and potential gene flow between and within the two species by estimating effective migration rates (Fig. 2b)[36]. Between-species comparisons revealed a general barrier through the Central African rainforest and the East African Rift Valley, separating W/C and E/S populations. Within-species comparisons revealed high connectivity within both bushpig and red river hog ranges respectively, with the exception of Malagasy and non-Malagasy bushpigs which exhibited a barrier across the Mozambique Channel, particularly with the northernmost non-Malagasy populations. A decrease in effective migration in Ethiopia was also observed; this is in contrast with Uganda where we observed weak barriers, suggesting a possible corridor of gene flow connectivity within this region.

To further investigate potential gene flow patterns between the two species, we then tested for ancient admixture events. $f$-branch statistics revealed putative signals of gene flow between bushpigs and red river hogs, exhibiting extensive gene flow involving Uganda (Supplementary Fig. 6). D-statistics were then used to specifically test whether there was increased allele sharing between red river hogs and bushpigs within the putative suture zone compared to populations further from the suture zone (e.g., Ghana and Madagascar). Two tests were designed to test this hypothesis. We first set the westernmost population (Ghana) as H1, each of the red river hogs as H2, and each of the bushpigs as H3 (Fig. 2c; upper panel). This revealed that all red river hog populations showed signs of gene flow from bushpigs, decreasing in signal from Central Africa to the westernmost red river hog populations, with a particularly strong signal from Uganda. Similarly, to test for gene flow in the opposite direction, we performed similar tests with the southernmost bushpig population, South Africa as H1, each of the remaining red river hog populations as H2, and each of the bushpig populations as H3 (Fig. 2c; lower panel). We observed a similar result, whereby we perceived a signal decrease towards more eastern and southern populations. Notably, this signal was particularly strong in Ethiopia and Uganda, suggesting substantial gene flow between red river hogs and these bushpig populations. Furthermore, we estimated the amount of gene flow between species using $f_4$-ratios, under assumptions that the populations considered had evolved together as a perfectly bifurcating tree and that the only gene flow event is the one that is modelled. This analysis estimated up to 21% gene flow from red river hogs into Uganda and up to 13% into Ethiopia, with increasing signals in more central populations (Supplementary Fig. 7). However, it must be cautioned that given the complicated history of these populations, as suggested by our $f$-branch ($f_b$) results (Supplementary Fig. 6), these values are unlikely to accurately represent historical gene flow events which may have occurred between multiple populations at different timepoints. Nevertheless, when considering all three analyses, these results suggest that there is or has been gene flow between the two taxa currently identified as species and that the gradient of allele sharing between them is consistent with isolation by distance, where genetic similarity is strongest in populations from Central Africa. Additionally, these results could also be interpreted as a complex network of populations connected by genetic exchange, either recent or ancient.

## Demographic histories and genetic diversity
Demographic histories of the surveyed populations were next explored (Fig. 3a)[37,38]. All PSMC curves overlapped from the most ancient past until ≈500 kya, where we observed a stark difference in PSMC trajectories between red river hog and bushpig individuals. All red river hog populations first experienced a moderate increase

(population expansion assuming panmixia) followed by a more recent contraction ≈50 kya. In contrast, bushpig individuals exhibited PSMC curves that followed three different trajectories: (i) the populations in Tanzania, Zimbabwe, South Africa, and Madagascar exhibited relatively constant (i.e., horizontal) curves until ≈10 kya; (ii) the Ugandan population showed a demographic history more similar to red river hogs than to the remaining bushpig populations, particularly between 100–500 kya, and; (iii) the Ethiopian population exhibited a history characterised by a declining and low PSMC curve ≈200 kya. These results, in combination with the D-statistics and $f_4$-ratio results reported above, suggest that the unique demographic histories in Uganda and Ethiopia could be influenced by their geographic location as a place of putative introgression between the two taxa.

Per-sample heterozygosity was next explored as a measure of genetic diversity, differing at both species and population levels (Fig. 3b). Heterozygosity was generally lower in bushpigs when compared to red river hogs, with the exception of Uganda and Tanzania which had similar heterozygosity levels to DR Congo and Eq Guinea. The bushpig population in Ethiopia exhibited extremely low genetic diversity, one third of that of the highest, Tanzania (Fig. 3b). This was consistent with elevated $F_{ST}$ values, reduced connectivity in EEMS, and the low effective population size as estimated by PSMC (Fig. 2a). Heterozygosity in Madagascar was also relatively low, but similar to that of Zimbabwe and South Africa.

Runs of homozygosity (ROH) were then explored using PLINK-based analyses within medium-high depth genomes (Methods), where the fraction of ROH with length >1 Mb ($F_{ROH}$) was found to affect 3–27% of the genome across all individuals (Fig. 3c). There was no systematic difference in $F_{ROH}$ between red river hogs and bushpigs. Additionally, we explored ROH in all 67 samples, including low-depth samples with ROHan[39], which overall yielded similar results. However, due to its window-based approach, ROHan could not identify most of the shorter ROHs (<2 Mb) and therefore the overall $F_{ROH}$ was slightly smaller using this approach (Supplementary Fig. 8). When considering both analyses, we observed heterogeneous levels of ROH within Uganda, Ethiopia, Cameroon, and Eq Guinea likely driven by recent inbreeding events, leading to differences in longer ROHs. Madagascar individuals had the largest $F_{ROH}$ out of all individuals tested, exhibiting the largest proportions for each length class <10 Mb for all samples excluding Ghana, and <5 Mb for all samples. This was consistent with results comparing linkage disequilibrium (LD) decay between different populations with at least five unrelated individuals, where Madagascar exhibited increased LD (Fig. 3d).

Taken together, these results suggest that the evolutionary histories of red river hogs and bushpigs vary markedly. In light of previous results, we find further evidence that Uganda is likely a region of strong introgression and that the Ethiopian population underwent strong drift after gene flow with red river hogs. Finally, we find that Malagasy individuals had similar population histories and a level of genetic diversity comparable with other southern bushpig populations, but had increased $F_{ROH}$ and LD.

## Bushpig arrival in Madagascar coincides with the expansion of Bantu speakers
The timing of arrival and geographical origin of bushpigs in Madagascar is still unresolved as previous lines of evidence, e.g., estimated split times and fossil records, appear to be contradictory. We therefore explored the putative founding of this population. We first measured the amount of shared history between the Malagasy population and each of the other populations (Fig. 4a)[40]. Our outgroup $f_3$ results suggest that South Africa and Zimbabwe had the longest shared history with Madagascar amongst sampled populations. This was consistent with our results exploring gene flow and connectivity, which exhibited a weaker barrier between Madagascar and these two southern populations when compared with other

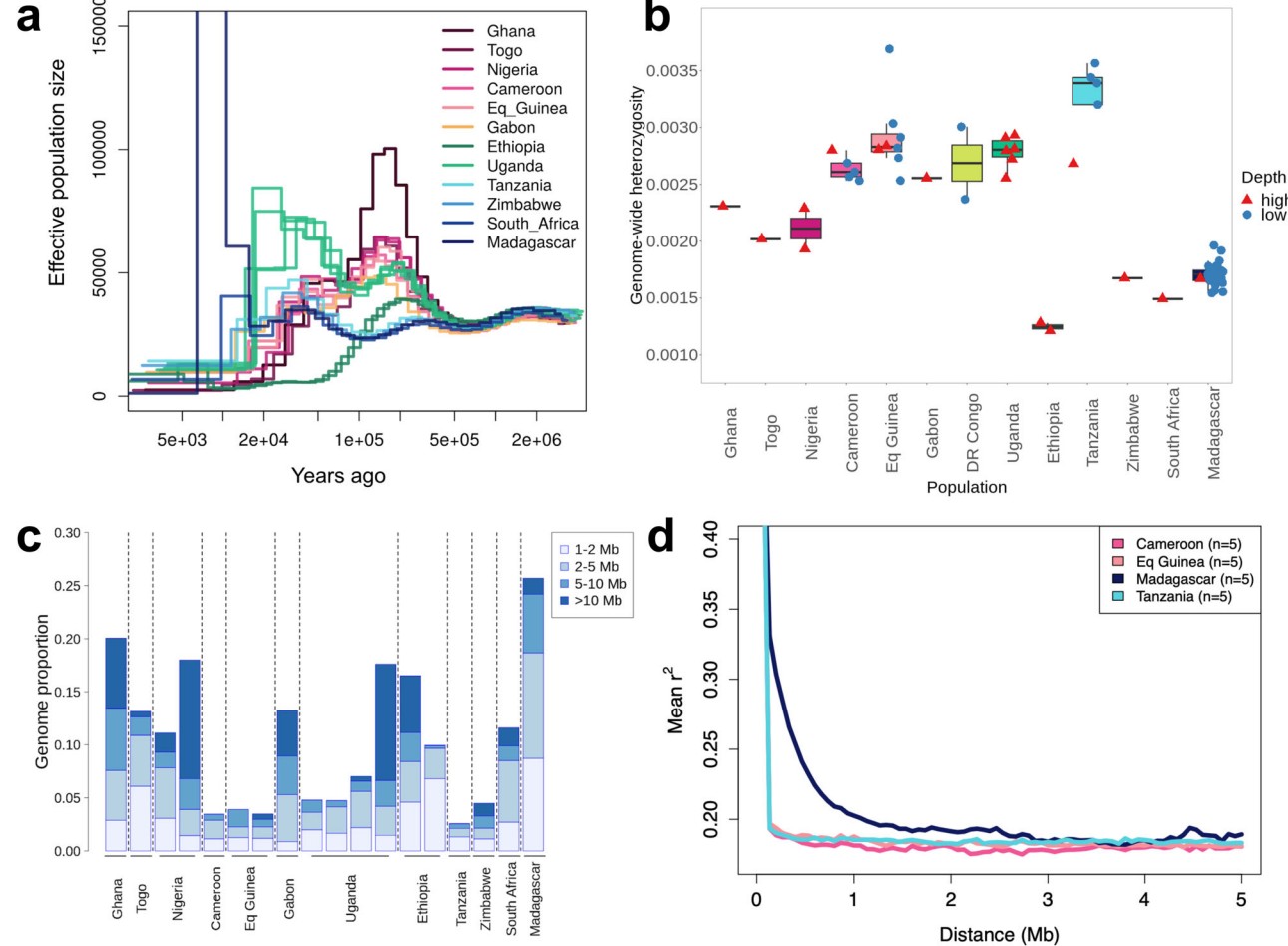

**Fig. 3 | Genetic diversity of African pigs. a** Effective population sizes over time for 18 medium-high depth pig samples as estimated by PSMC, assuming a mutation rate of $\mu = 1.49e^{-8}$ per site per generation and a generation time of six years[4,84]. **b** Genome-wide heterozygosity measurements described as the proportion of heterozygous sites per bp across each individual genome. Medium-high depth (≥14×; $n = 18$; red triangles) and low-depth samples ($n = 49$; blue circles) are shown. Boxplots indicate median (centre line), 25th and 75th percentiles (box), and the highest and lowest values within the upper and lower quartiles ± 1.5* interquartile range, respectively (whiskers). **c** Estimated genome-wide runs of homozygosity (ROH) proportions for 18 medium-high depth individuals. Each bar represents a single individual, grouped by their population. Proportions of differing ROH length intervals are shown as subdivisions within bars. **d** Linkage disequilibrium decay for populations with five or more samples, described as mean $r^2$ values for SNP pairs 0–5 Mb apart ($n = 5$ for each population).

bushpig populations (Fig. 2b, c), the neighbour-joining tree (Fig. 1c), and $D_{xy}$ values (Fig. 2a).

Split times between populations were next estimated using the Two-Two method (TT)[41,42], including the species split between red river hogs and bushpigs and the split between bushpig populations on mainland Africa and Madagascar (Fig. 4b). The species split was estimated to have occurred ≈300 kya. Consistent with the outgroup $f_3$ results, Madagascar exhibited the lowest split times with populations in South Africa and Zimbabwe (≈850 and ≈500 years ago, respectively). This further suggests that either one of these or an unsampled population within the same geographic region was the population of origin. Additionally, we investigated putative recent demographic events for the Madagascar population (Fig. 4c)[43]. This analysis suggested that the Malagasy population experienced a severe bottleneck, likely a result of a founder event between 1 and 5 kya. This result was also consistent with the high $F_{ROH}$ (Fig. 3c) and the high LD (Fig. 3d) characterising this population.

## Discussion
Biodiversity patterns on the African continent show striking similarities across multiple species, including a division between lineages in W/C Africa and in E/S Africa, with known hybridisation zones spanning

across Uganda, South Sudan, and Ethiopia[10,12,13,44]. Although hybridisation between the red river hog and the bushpig has been suggested before, this has not yet been studied in detail and the level of evolutionary divergence between them remains contentious. Moreover, bushpigs also represent the anomaly of being the only large wild mammal which occurs both on mainland Africa and Madagascar, but due to limited data from Malagasy bushpigs, the scenario of colonisation is still largely unknown. Our study investigates the evolutionary histories of red river hogs and bushpigs at the genome scale, allowing for a better understanding of the processes leading to the formation of two distinct taxa and the colonisation of Madagascar.

### Divergence and introgression between red river hogs and bushpigs
Under simplifying assumptions, we found that split times between red river hogs and bushpigs could be estimated at ≈300 kya (Fig. 4b) and that this time was in the same range as W/C-E/S split time estimates for other African species which exhibit a similar phylogeographic pattern, including African buffalo (≈273 kya)[10], baboon (≈320 kya)[13], giraffe (≈300 kya)[45], warthog (≈226 kya)[4], lion (≈245 kya)[11], and spotted hyena (≈360 kya)[46]. Although red river hogs and bushpigs are widely considered to be distinct species, in the examples mentioned above, the

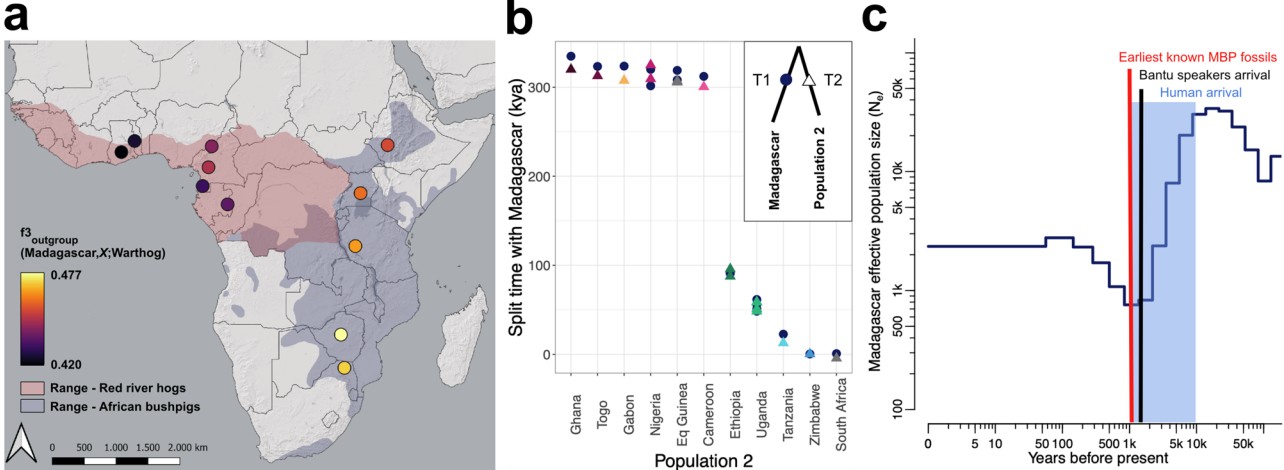

**Fig. 4 | Origin and timing of bushpigs in Madagascar. a** Outgroup $f_3$ statistics in the form $f_3$ (Madagascar, $X$; Warthog), where $X$ describes different sampling localities. **b** Population split times with Madagascar as estimated by TT using individual pairs of medium-high depth (≥14×) samples ($n = 18$). T1 and T2 values, describing population split times from a common ancestor, are shown as dark blue circles for Madagascar and coloured triangles for other populations (Population 2), respectively. A mutation rate of $\mu = 1.49e^{-8}$ per site per generation and a generation time of six years were assumed[4,84]. kya – thousands of years ago. **c** Recent effective population sizes inferred based on unrelated Madagascar individuals using popSizeABC[43] ($n = 21$). Shaded region – estimated timing of human arrival in Madagascar[24,26,27]; black line – estimated timing of Bantu speakers arrival in Madagascar as estimated in Pierron et al.[26]; red line – estimated timing of earliest known bushpig fossils in Madagascar (MBP)[63].

divergent populations are typically considered to belong to the same species, except for baboons, elephants, and with an ongoing taxonomic debate about the species status of giraffes[12,45,47,48]. In a previous study, Gongora et al.[3] estimated the divergence time between red river hogs and bushpigs at 2710 kya, thus lending support for a species distinction between red river hogs and bushpigs. However, this estimate was obtained with a wide confidence interval of 200–4800 kya[3]. Our results suggest (and corroborate recent findings) that divergence times between African suid taxa have thus far been overestimated[4]. This includes red river hogs and bushpigs, where our analyses represent a much younger divergence time and more reticulated evolutionary history than what was previously known.

Furthermore, our results indicate a complex history of population structure with possible periods of increased and decreased connectivity between populations. PSMC curves can be interpreted as representing changes in coalescent effective population size, as is usually done, but this interpretation relies on a very strong assumption of total panmixia[49]. If this assumption is violated, changes in PSMC curves may alternatively reflect changes in gene flow[50,51]. Thus, an alternative explanation for the observed PSMC curves is that there was a major fragmentation period between 500 kya and 100–200 kya, and a second period more recently, possibly between 100 and 30 kya. Such complex histories could lead to overestimates of divergence times.

Our findings, therefore, have implications for the ongoing taxonomic debate about *Potamochoerus* and for the interpretation of genetic data in this group. The current taxonomic definition of *P. porcus* and *P. larvatus* was primarily based on morphology and a lack of 'convincing evidence' that they interbred when they came into contact[8], a case which has been previously disputed by authors who favour a single-species taxonomy[52,53]. Our results suggest that the two taxa could be a case of incomplete speciation. However, we emphasise that in cases such as *Potamochoerus*, different species concepts might arrive at different conclusions about whether speciation has gone to completion and note that taxonomic revisions should draw on various types of data and evidence, e.g., morphology and behaviour[54], which were not considered in the present study.

The impact of changing climate and habitat availability on the complex evolutionary history between red river hog and bushpig is showcased by the Ethiopian population, which has received substantial red river hog gene flow and was fixed for a divergent red river hog mtDNA lineage (Supplementary Fig. 9). Ethiopia is further characterised by a low effective population size and is strongly affected by drift, as illustrated by relatively long ROHs and high $F_{ST}$ values. Hence, Ethiopian bushpigs show contrasting evidence of connectivity and isolation, possibly caused by historical fluctuations in the equatorial forest belt across Africa. These fluctuations could have facilitated intermittent contact and hybridisation between red river hogs and bushpigs as the forest expanded, followed by isolation of resident populations as the forests receded. In line with this, the taxonomic status of *Potamochoerus* in Ethiopia remains unresolved[55] and there is anecdotal evidence of African buffalos in Ethiopia that strongly resemble the forest buffalos of distant Central and West Africa[56]. Similarly, the observation of *Potamochoerus* admixture in Uganda coincides with the present-day boundary between two of Africa's major mammalian biogeographical regions, the Guinean-Congolian and the Sudanian core regions[57], an area which also constitutes well-known hybridisation zones for several large-mammal taxa, including elephants[58], and subspecies of buffalo[10] and kob (*Kobus kob*)[59]. However, without more samples from adjacent locations such as East DR Congo, the centre of hybridisation is speculative.

### Origin of Malagasy bushpigs

Our results support the hypothesis that bushpigs were introduced by humans from southeastern Africa into Madagascar 1000–1500 years ago[25,26] and possibly as early as 5000 years ago. A previous estimate for the most recent common ancestor of mainland and Malagasy bushpigs at 480 kya[29] contradicts this; however, this could be partially caused by problematic temporal calibration and by the limited information contained in mtDNA sequences. Although we cannot pinpoint the precise source population from which the Malagasy bushpigs were introduced with certainty, our results suggest an origin in southern Africa, as corroborated by the Zimbabwe and South African populations being closer than all other populations when using NGSadmix, evolutionary distances, outgroup $f_3$ statistics, divergence times, and EEMS. Our popSizeABC estimate of an effective population size of 1000 individuals during the bottleneck 1500 years ago is surprisingly high, assuming that the founder event was a single occurrence involving a limited number of individuals carried to Madagascar by ship.

The estimate is supported by heterozygosity levels similar to those observed in southern Africa, although we cannot know to what extent southern African populations were subject to drift since the founding of the Malagasy population. Multiple introductions, spanning over a longer period of time or with animals sourced from different mainland populations (including already admixed individuals), however, may have inflated this estimate. Furthermore, an instant large drop in population size followed by subsequent regrowth will be estimated by popSizeABC as a more gradual decline that starts earlier[43], making it difficult to pinpoint the exact timing of when the bottleneck started.

Despite these limitations, our results provide, to our knowledge, the clearest evidence yet of a recent introduction of bushpigs to Madagascar mediated by humans, most likely through populations that started to arrive on Madagascar from Southeast Africa at least 1500 years ago and possibly earlier[26], although the latter has been contested[60]. Triangular Incised Ware (Tana Ware) style pottery remains dated to 1000–1400 years ago were found in human settlements in southern Madagascar, suggesting either contact with or colonisation from the African Swahili region at this time[61]. Bushpigs likely became established on the island as livestock together with zebu, goats, and sheep 700–1200 years ago based on $^{14}C$ bone analyses[28], coinciding with the extinction of Madagascar's megafauna as the result of hunting, pastoralism, and farming[62]. Lending strong support to this hypothesis, our dating results are in line with the oldest fossils of Malagasy bushpigs (≈1000 years ago)[63]. Note that even if bushpigs arrived in Madagascar around 1500 years ago, it does not refute that humans arrived earlier, but does suggest that a significant human migration event occurred at this time. Altogether, our results are consistent with Blench's[22] hypothesis that human migrants reaching Madagascar captured bushpigs in Southeast Africa, introduced them to Madagascar, and made an attempt to domesticate them. Etymological problems over the naming of Malagasy bushpigs (i.e., with a term usually used for bovine in Southeast Asia) highlight that there are still outstanding questions regarding the cultural perception and uses of bushpigs in early Malagasy settlers, composed of both Bantu-speaking and Austronesian-speaking people. Furthermore, the alleged morphological variation between Malagasy bushpig subpopulations[5], including the suggestion that they are distinct subspecies[8], had led to suggestions of multiple distinct introduction pulses through the Comoros Islands and the North Mozambique current[64]. Although we were unable to explicitly test this hypothesis without samples from the Comoros, our results from the PCA, NGSadmix, and IBS tree do not suggest multiple pulses into Madagascar and did not identify substantial structure within the island, which is consistent with a relatively homogeneous founder population.

Though we have samples across most of the species' ranges, we acknowledge that there are gaps in our inferences. A more even spread across the species ranges and especially more sampling localities from within the putative suture zone, e.g., southern DR Congo, would increase our understanding of the evolutionary dynamics near the suture zone. In addition, more sampling localities along the East African coastline could help to identify more precisely, the source populations for the colonisation of Madagascar[5].

Overall, our study sheds light on the distribution of genomic diversity and the evolutionary histories of two closely related African pig taxa. It provides yet another example of diverged taxa with a suture zone around western Uganda, as has been shown for numerous other taxa and is characteristic for African mammal phylogeography. The recent split times, moderate and large values of $F_{ST}$ as geographic distance increases, and ancestral gene flow between bushpigs and red river hogs suggest that their evolutionary divergence is young and incomplete, a perspective that should be taken into account in future taxonomic assessments and management plans. Furthermore, our data from Malagasy bushpigs suggest that bushpigs indeed colonised the island by hitchhiking along with the accelerating human

colonisation of Madagascar occurring around the onset of the Medieval period. These insights provide answers for long-standing questions regarding the distribution of biodiversity in Africa and the mysterious presence of the bushpig in Madagascar.

## Methods

### Sample collection and laboratory protocol
The research presented in this study complies with all relevant ethical regulations. The research was carried out in compliance with the Code of Conduct for Responsible Research at the University of Copenhagen. Samples were sourced from existing scientific collections, as detailed in Supplementary Data 1; no destructive sampling was performed as part of this study. Samples were treated as follows: for non-USDA samples, the QIAGEN DNeasy Blood & Tissue Kit (QIAGEN, Valencia, CA, USA) was used for DNA extraction following the manufacturer's protocol. RNase was added to all samples to ensure RNA-free genomic DNA. DNA concentrations were then measured using a Qubit 2.0 Fluorometer and Nanodrop before using gel electrophoresis to check the quality of genomic DNA. Bushpig hide samples contributed by the USDA were salted, acidified, and dried after collection in the field. Samples were purchased by the USDA from willing sellers and stored at −20 °C until DNA extraction by standard phenol/chloroform procedures. DNA was dissolved in a solution of 10 mM Tris-HCl, 1 mM EDTA (TE, pH 8.0) and stored at 4 °C. Sample quality and concentrations were measured by ultraviolet spectrophotometry and double-stranded DNA fluorometry (DeNovix Inc., Wilmington, DE USA; QuantiFluoONE, Promega, Madison, WI, USA).

### Sequencing and mapping
All samples were sequenced using Illumina paired-end 150 bp reads. This included 53 samples which were sequenced to low depth (≈3–6× depth of coverage) on the Illumina NovaSeq platform and 16 samples sequenced to medium-high depth (≈14–49×) on the Illumina HiSeq2500 and NextSeq2000 platforms (Illumina Inc., San Diego, CA, USA). Sequencing data were assessed using FastQC v0.11.9[65] and MultiQC v1.13[66]. Publicly available data from two red river hog samples from Nigeria were also used in this study (SAMC146518; SAMC146529) (Supplementary Data 1)[67]. As downstream analyses such as PSMC exhibit high variance when using lower-coverage samples (≤10×)[68], we utilised a threshold of >12× depth of coverage to classify 18 unrelated individuals (≥14×) as medium-high depth samples within this study (Supplementary Data 1).

The sequencing reads were mapped to the chromosome-level assemblies for the common warthog (*Phacochoerus africanus*, accession number: GCA_016906955.1) using a development version of the PALEOMIX BAM pipeline[69] (https://github.com/MikkelSchubert/paleomix; branch 'pub/2022/africa').

Reads were processed using AdapterRemoval v2.3.2[69] to remove adapter contamination and to merge overlapping reads in order to improve read fidelity. Adapter sequences published by Illumina and BGI were used for trimming. Reads were merged using the --collapse-conservatively option, which assigns 'N' to any mismatching position in the alignment for which both bases have the same quality. No trimming of Ns or low-quality bases was performed and only empty reads resulting from primer-dimers were excluded. Trimmed reads were subsequently mapped using BWA-mem v0.7.17-r1188[70]. Reads were post-processed using samtools v1.11[71] commands 'sort' and 'calmd' and putative PCR duplicates flagged using the 'markdup' command and PALEOMIX 'rmdup_collpased', for paired and unmerged reads respectively.

The resulting BAM alignments were filtered to remove unmapped reads, secondary alignments, PCR duplicates, and supplementary alignments, and reads flagged as having failed QC. We furthermore removed alignments with an inferred insert size <50 bp or >1000 bp and reads where less than 50 bp or 50% were mapped to the reference

genome. Finally, we removed pairs of reads mapping to different contigs or in an unexpected orientation and reads for which the mate had been removed by any of the above criteria.

## Genotype calling

Based on the mapped reads, we used bcftools v1.13[71] to call genotypes for the 18 medium-high depth individuals and three outgroup samples; common warthog (*Ph. africanus*), desert warthog (*Ph. aethiopicus*) (Supplementary Data 1), and domestic pig (*Sus scrofa*; accession: SAMN28197093), requiring minimum base quality 25, mapping quality 30, and ignoring contigs smaller than 100 kb. We used the '*-per-sample-mF*' flag for the pileup, and the '*--multiallelic-caller*' for the calling.

From the initial calls, we filtered out sites based on mappability, heterozygosity, and depth outliers as described in the following section and removed all indels and multiallelic sites. Finally, any genotype call with less than 10 read depth was set as missing, along with heterozygous calls with less than two reads supporting the minor allele.

## Reference genome and site quality filters

We estimated the mappability of the warthog reference genome using GENMAP v1.3.0[72]. Here, we used 100 bp k-mers allowing for two mismatches (*-K* 100 *-E* 2) and the remaining parameters set to default settings. All sites with a mappability score <1 were excluded from downstream analyses. RepeatMasker v4.1.1[73] was used to identify repeat elements in the warthog genome assembly, utilising '*rmblast*' as the search engine and '*mammal*' as the query species with default settings. Repeat regions identified with RepeatMasker were masked to limit mismapping in these regions. Annotated sex chromosomes and scaffolds that were not assembled into chromosomes were also excluded.

We also removed genomic regions with unusually high heterozygosity to avoid mismapping artefacts driven by multimapping on paralogous and other repetitive regions. We first estimated genotype likelihoods for SNPs using Angsd v0.925[74] with the GATK model (*-GL* 2), minimum mapping quality of 30 (*-minMapQ* 30), a minimum base quality of 30 (*-minQ* 30), a *p*-value of 1e$^{-6}$ to call SNPs (*-snp_pval* 1e$^{-6}$) and kept only SNPs with minor allele frequency (MAF) > 0.05 (*-minmaf* 0.05). Genotype likelihoods were then used as input for PCAngsd's per site Hardy-Weinberg equilibrium (HWE) test[75], which estimates inbreeding coefficients (*F*), and a likelihood ratio test statistic (LRT) for evidence of deviation from HWE, while controlling for population structure. The PCAngsd MAP test[75] was also used to select the optimal number of principal components in each case. Sites with *F* < −0.9 and LRT > 24 were subsequently removed as they may have been driven by mapping artefacts, and therefore all regions within 10 kb from such sites were also discarded. We ran this analysis separately for red river hog and bushpig samples.

Finally, we removed sites with extreme depth. We estimated the global depth (read count across all samples) for each site using Angsd[74] (*-minMapQ* 30 *-minQ* 30 *-doCounts* 1 *-doDepth* 1 *-dumpCounts* 1 *-maxdepth* 4000). This was done separately for each species for all (*n* = 67), unrelated (*n* = 54) and medium-high depth samples (*n* = 18). Only autosomal chromosomes were included. From the global depth we calculated the upper 1% and lower 3% percentiles and visually inspected the plots before deciding on a threshold for excluding sites with extreme sequencing depth. Only sites that were within the thresholds for both low- and medium-high depth samples were used in the downstream analyses.

## Sample filters

We identified and excluded samples with high sequencing error rates based on the "perfect individual" approach[76]. The rationale behind this approach is that any sample in the dataset should have equal genetic distance to the outgroup and therefore samples with excess/deficit of derived alleles would be interpreted as errors. As the "perfect

individual" we used a high depth individual from Ghana (BPig-Gha0038). This sample was processed with Angsd[74] to create a consensus sequence (*-doFasta* 2), taking the most commonly observed base as the consensus (*-doAncError* 2) while setting the base quality to at least 30 (*-minQ* 30). We chose the common warthog as an outgroup and mapped all samples to the consensus using BWA excluding sex chromosomes, the mitogenome, repeats, and sites with mappability <1. Individuals with high error rates (> 0.001) were removed from downstream analyses (Supplementary Fig. 1).

We then considered relatedness between samples, where we identified and removed potential relatives and duplicated samples using the methodology described in IBSRELATE[77]. First, we calculated the Site Allele Frequency (SAF) likelihood in Angsd[74] for each individual. We used the genotype likelihood-based approach assuming HWE (*-doSaf* 1). The warthog genome was used as ancestral reference (*-anc*), a minimum mapping quality of 30 (*-minMapQ* 30), a minimum base quality of 30 (*-minQ* 30), and the GATK method (*-GL* 2). Then, we inferred the two-dimensional site frequency spectra (2D-SFS) pairwise among all possible combinations of individuals. To limit computational time, we limited the number of sites surveyed to the first 50,000 sites. Based on the 2D-SFS, we calculated R0, R1 and KING-robust kinship[77,78], which can be used to identify close familiar relatives. For the analysis, we combined all the data from bushpigs and red river hogs in this analysis to account for potentially interspecies duplicates or mislabelled samples. We identified and removed an individual from each pair of first- and second-degree relatives.

## Imputation

Imputation was performed using BEAGLE 3[79] from genotype likelihoods (GLs) estimated in Angsd[74]. GLs were estimated using the GATK genotype likelihood model (*-GL* 2) and only keeping sites that had a *p*-value less than 1e$^{-6}$ (*-SNP_pval* 1e-6) for being variable in addition to only keeping sites that passed initial QC (*-sites*) as well as using a minimum MAF of 0.025 (*-minMAF* 0.025). We assumed the major allele was fixed and the minor was unknown when estimating GLs (*-doMajorMinor* 1 *-doMAF* 2). We further filtered imputation results by only keeping sites with an imputation score $R^2$ > 0.95 and which had a maximum of 5% missingness after applying a >0.95 posterior probability cutoff on genotype calls.

## PCA, IBS and population structure

Beagle GL input files were first generated using Angsd[74], keeping only the sites that passed QC, with additional filters of removing tri-allelic sites, and with a minor allele frequency filter of 0.05. We used PCAngsd v1.02[75] to estimate the covariance matrix and identify potentially population structure for all individuals. A pairwise identity-by-state (IBS) matrix was then generated using Angsd, using the sample filters and including the *-doIBS* 1 flag. A neighbour-joining tree was then estimated using this matrix using the *ape* library in R[80].

## NGSadmix & evalAdmix

Admixture proportions for each population were inferred based on GL using NGSadmix[32]. A Beagle file, using the same filters to investigate population structure with PCAngsd, was taken and randomly thinned to contain one million sites for computational practicality. We ran NGSadmix with *K* = 2 to *K* = 9 until the model converged, where the top 3 maximum likelihood runs were within 10 log-likelihood units of each other or until a limit of 4000 independent runs was reached without convergence. *K* = 9 did not converge after 4000 independent runs, likely constrained by the number of samples per population. Model-based analyses of population structure make a set of assumptions about the data (e.g., individuals are unrelated, are in HWE, exhibit no LD, and that each ancestral population is represented by multiple unadmixed individuals with no subsequent drift). Therefore, we calculated the correlations of residuals using evalAdmix[33] for each pair of

individuals to evaluate model fit and to test whether the data violated some of these assumptions for $K$ ancestral clusters.

## Population differentiation ($F_{ST}$ / $D_{xy}$)

To quantify the extent of genetic differentiation between red river hog and bushpig populations, we used Hudson's estimator for genome-wide $F_{ST}$[34]. This analysis encompassed two approaches: one utilising called genotypes for the 18 medium-high depth genomes (Supplementary Fig. 5), and another utilising all 54 unrelated genomes and estimating values from population-level 2D-SFS inferred from genotype likelihoods using winsfs v0.7.0 (https://github.com/malthesr/winsfs). We also calculated the absolute genome-wide nucleotide divergence ($D_{xy}$) for all population pairs with 2D-SFS using a custom Python script (https://github.com/ivanliu3/asfsp)[81].

## Estimation of effective migration surfaces (EEMS)

To investigate effective migration and gene flow connectivity between populations, we used the Estimated Effective Migration Surfaces (EEMS) program[36]. A distance matrix was created from individual-level 2D-SFS estimated from GLs and was used as input for the program. EEMS v0.0.0.9000 was run using 300 demes for three independent runs of 30 million iterations, discarding the first 15 million as burn-in. Convergence was assessed visually and by using the Gelman-Rubin diagnostic in the *coda* R package[82] and generated using the *reemsplot* package (http://www.github.com/dipetkov/eems).

## D- and *f*-branch statistics

To explore signatures of introgression between red river hog and bushpig populations, the Dsuite package v0.5-r44[83] was utilised on variable sites of medium-high depth individuals as input, with the topology of a neighbour-joining tree based on pairwise Hudson's $F_{ST}$ between individual pairs using the *ape* library in R[34,80] and the common warthog as an outgroup. The Dtrios function in Dsuite calculates the D-statistics for all possible trio combinations, which are then used for calculating *f*-branch statistics, using the *f*-branch command. A summary of these results within the provided phylogenetic framework is presented as a heatmap (Supplementary Fig. 6).

## *f*₄-ratio estimates

$f_4$-ratios were calculated using ADMIXTOOLS v7.0.2[40] with genotype calls from medium-high depth individuals as input. $f_2$ statistics were first calculated for each population using five million bp blocks, as for outgroup $f_3$ estimations (Methods). Using the common warthog as an outgroup, $f_4$-ratios were then estimated in the form $f_4$ (A,O;X,C)/$f_4$ (A,O;B,C). $f_4$-ratios were calculated for three different scenarios: (a) where we estimated admixture proportions into Uganda from non-Ghanian red river hog populations, (b) where we estimated admixture proportions into Ethiopia from non-Ghanian red river hog populations, and (c) where we estimated admixture proportions into each of the bushpig populations from Eq Guinea red river hogs. We note that these analyses carry assumptions that populations considered had evolved as a perfectly bifurcating tree and that the only gene flow event is the one that is modelled. As suggested by our *f*-branch ($f_b$) results (Supplementary Fig. 6), the relationship between the populations is likely much more complicated and therefore one should be cautious in interpreting these results as evidence for a single point migration event. The amount of gene flow could be much higher from an unsampled population, or it could be many migration events happening between several populations at different timepoints.

## PSMC

The Pairwise Sequentially Markovian Coalescent (PSMC) algorithm[37,38] was used to infer changes in historical population sizes for all individuals sequenced at medium-high depth. PSMC v0.6.5 was run with default parameters. In addition to the size quality filter, we also excluded sites based on the average depth per individual divided by three as a minimum and twice the average depth per individual as a maximum. We used a mutation rate of $1.49e^{-8}$ per site per generation and a generation time of six years, as described for warthogs[4,84].

## Heterozygosity

Genetic diversity of pig populations was approximated through the estimation of genome-wide heterozygosity. Individual-level heterozygosity was estimated in Angsd[74] using individual-level site frequency spectra, measured as the proportion of heterozygous loci per sample. The GATK genotype likelihood model was utilised in Angsd (-*GL* 2), with minimum quality filters on mapping (-*minMapQ* 30) and base quality (-*minQ* 30), while reducing the amount of reads with excessive mismatches (-*C* 50).

## Runs of homozygosity

Runs of homozygosity (ROH) analyses were performed using PLINK v1.9[85]. PLINK files included only filtered variable sites within medium-high depth samples ($n = 18$), with an additional depth filter (10 reads minimum) and at least two reads carrying each of the two alleles at heterozygous sites. In order to generate more accurate ROH regions, we further excluded SNPs with MAF < 0.05 (-*-maf*) and missing genotype calls (-*-geno*) <0.05. For each individual, we then used PLINK with -*-homozyg* to scan the ROH regions, with scanning window modifiers (-*-homozyg-window-het* 3 -*-homozyg-window-missing* 20). A few SNP sites with >50% heterozygous genotypes across individuals were also excluded since these were prone to genotype errors which can break up longer ROHs.

In addition, ROHan v1.0.1[39] was used to estimate local rates of heterozygosity and infer runs of homozygosity for all 67 samples after site filtering ('Reference genome and site quality filters'; Methods). The hmm was run with default numbers of steps (1000) and chains (50000) and with −*rohmu* 1e⁻⁴. ROHs inferred from mid estimates of heterozygosity were subsequently used for downstream analyses (mid.hmmrohl.gz). Analyses were parallelised using GNU Parallel[86].

## LD decay

Linkage disequilibrium (LD) decay curves were generated for four populations which included at least five samples (Cameroon, Eq Guinea, Madagascar, and Tanzania) to reduce the potential bias among the variable sample sizes among populations[87]. We calculated LD using the *relate* R package[88] for each population using imputed polymorphic sites from chr16. These sites were thinned to 10% of the original data using PLINK v1.9 (-*-thin* 0.1) function[85] to minimise computational time. Pairwise LD was calculated using 36,417 SNPs and 5 Mb physical distance, at which the curves plateaued.

## Outgroup $f_3$ statistics

To further test gene flow between the Malagasy population and other red river hog and bushpig populations, outgroup $f_3$ statistics were calculated based on genotype calls from medium-high depth individuals using ADMIXTOOLS v7.0.2[40]. $f_2$ statistics were first calculated for each population using five million bp blocks. Using the common warthog as an outgroup, outgroup $f_3$ was estimated in the form $f_3$ (Madagascar, $X$; Warthog), where $X$ represents different populations of red river hogs and bushpigs.

## TT and split time estimations

Population split times were estimated using the Two-Two (TT) method[41,42]. This approach estimates separate split times from a common ancestor for two populations and has been used in various studies, including human[42,89-91], direct ancestry[92], and animal studies[4,93]. T1 describes the estimated time to the common ancestor for population 1 and T2 the time to the common ancestor for population 2. Further details for the method are described in Sjödin et al.[41]

and its relationship with other similar methods in Mualim et al.[94]. For this analysis, we utilised unfolded 2D-SFS from medium-high depth individuals polarised using three outgroups: against the common warthog (*Phacochoerus africanus*), desert warthog (*Ph. aethiopicus*) (Supplementary Data 1), and the domestic pig (*Sus scrofa*; accession: SAMN28197093). A mutation rate of $1.49e^{-8}$ per site per generation and a generation time of six years were used to scale split times as in PSMC analyses[4,84].

## PopSizeABC

In order to estimate recent population size changes that cannot be captured by PSMC, we used popSizeABC[43] on imputed data, with a focus on a sufficiently large sample of unrelated individuals ($n = 21$) in the Malagasy bushpig population. PopSizeABC takes VCF files per chromosome as input and estimates linkage disequilibrium curves and site frequency spectra for tested populations. PopSizeABC population size estimates require multiple simulations of demographic scenarios to compare a posterior distribution of simulation derived parameters to those observed in the real data. For this analysis, 210,000 simulations were performed for 100 2 Mb regions per simulation as per the suggested settings in the popSizeABC publication for the software. A minimum MAF threshold of 0.1 was applied to calculation of the site frequency spectra and 0.2 for calculation of the linkage disequilibrium curves, again in accordance with the suggested parameters in Boitard et al.[43]. The same recombination rate, mutation rate and generation time used in PSMC and TT were used for popSizeABC.

## Mitochondrial DNA phylogeny

Bioinformatic analyses were performed with the Geneious v2023.0.1 (https://www.geneious.com/) package. First, we trimmed and quality-filtered the reads with BBDuk v37.64 of the BBTools package (https://jgi.doe.gov/data-and-tools/bbtools/) and removed duplicate reads with Dedupe v37.64 (BBTools package), applying standard settings for both filtering steps. We then mapped the reads to the *P. porcus* reference mitogenome (NC_020737) using the Geneious mapper with standard settings (medium-low sensitivity, fine-tuning with up to 5 iterations), annotated the mitogenome using the reference sequence as template, and checked the results by eye. Variant calling was verified with the find variations/SNPs tool in Geneious with standard settings (maximum variant $p$-value $10^{-6}$, minimum strand bias $p$-value $10^{-5}$ when exceeding 65% bias).

For phylogenetic reconstructions, we removed identical haplotypes (Supplementary Data 4) and added additional mitogenome sequences available in GenBank (*P. porcus*: NC_020737, *Ph. africanus*: NC_008830, *Porcula salvania*: NC_043879, *Sus scrofa*: NC_000845, *S. cebifrons*: NC_023541, *S. celebensis*: NC_024860, *S. barbatus*: NC_026992, *S. verrucosus*: NC_023536). We aligned the 52 sequences in the final alignment with MUSCLE v3.8.31[95] in AliView v1.18[96] and corrected the alignment by eye. We reconstructed a phylogenetic tree with the maximum-likelihood (ML) algorithm using IQ-TREE v2.2.0[97]. Therefore, we treated the mitogenome as a single partition and applied the optimal substitution model (GTR + I + G) as determined with ModelFinder[98,99] in IQ-TREE under the Bayesian Information Criterion (BIC). To obtain node support for the ML analysis, we performed 10,000 ultrafast bootstrap (BS) replications[100].

We calculated divergence times in BEAST v2.6.7[101] applying a relaxed lognormal clock model of lineage variation[102], a Birth Death tree prior, and the best-fit model of sequence evolution as selected by ModelFinder. To calibrate the molecular clock, we set an age constraint on the node of the most recent common ancestor (MRCA) of Suinae with a mean of 10.0 million years ago (Mya) and a 95% highest posterior density of ±1.0 Mya[1,103]. We ran the analysis for 100 million generations with tree and parameter sampling setting-in every 5000 generations. To assess the adequacy of a 25% burn-in and convergence of all parameters, we inspected the trace of the parameters across generations using Tracer v1.6 (http://tree.bio.ed.ac.uk/software/tracer/). We combined sampling distributions of four independent replicates with LogCombiner and summarised trees with a 10% burn-in using TreeAnnotator (both programs are part of the BEAST package). We visualised all phylogenetic trees in FigTree v1.4.2 (http://tree.bio.ed.ac.uk/software/figtree/).

## Reporting summary

Further information on research design is available in the Nature Portfolio Reporting Summary linked to this article.

## Data availability

Data used in this study are described within the Article, Supplementary Data, and Supplementary Information. Sample locations and their sources are described within Supplementary Data 1. Raw fastq files generated in this study and their associated metadata are publicly available on the NCBI database under BioProject accession PRJNA1027560. BioSample accessions for each sample are described within Supplementary Data 1. Raw data from Xie et al.[67], available on the NGDC database (SAMC146518 and SAMC146529), a chromosome-level assembly of *Phacochoerus africanus*, available on the NCBI database (GCA_016906955.1), and reads from *Sus scrofa* (SAMN28197093) were also used in this study. Additional mitogenome sequences available in GenBank were utilised for mitochondrial analyses (*P. porcus*: NC_020737, *Ph. africanus*: NC_008830, *Porcula salvania*: NC_043879, *Sus scrofa*: NC_000845, *S. cebifrons*: NC_023541, *S. celebensis*: NC_024860, *S. barbatus*: NC_026992, *S. verrucosus*: NC_023536) Source data are provided with this paper.

## Code availability

Code used for analyses within this study are publicly available on Github: https://github.com/popgenDK/seqAfrica_bushpigs. A description of software and their version numbers can be found in Methods.

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

## Acknowledgements

We thank Amal Al-Chaer for her invaluable help with DNA extractions. We are also grateful to Peter Arctander, who organised sample collections between the 1980s and 1990s and to David Moyer for contributing Tanzanian bushpig samples, collected between 1995 and 1997 and subsequently stored in the collection at the University of Copenhagen. We would also like to acknowledge our collaborators from African wildlife management authorities for granting express permission to use samples within this study. We thank Komlan Afiademanyo (Université de Lomé, Togo), Flobert Njikou (Université de Yaoundé I, Cameroon), Alain Didier Missoup (Université de Douala), Gabriel Ngua (ANDEGE, Equatorial Guinea), and Jonas Muhindo and Idriss Ayaya (CIFOR, Democratic Republic of Congo) for collecting red river hog samples in West and Central Africa. We are also grateful to Jonathan Kingdon for providing express permission for the use of his illustrations within this manuscript. RFB, XL, and IM are supported by a Villum Young Investigator grant (VIL19114) awarded to IM. A. Albrechtsen, RFB, LL, and ZL are funded by the Novo Nordisk Foundation (NNF20OC0061343). ABO is supported by a Carlsberg Foundation Reintegration Fellowship (CF19-0427). MSS is supported by a Carlsberg Foundation Reintegration Fellowship (CF20-0355). MSR and A. Albrechtsen are supported by the Independent Research Fund Denmark (8021-00360B). GGE and RH are supported by a Danmarks Frie Forskningsfond Sapere Aude research grant (DFF8049-00098B), and RH is further supported by a Carlsberg Young Researcher grant (CF21-0497). A. Atickem is supported with a Rufford Small Grants for Nature Conservation (1118-C). GB, JS, LC, and PG are members of the EDB laboratory, which is supported by the Laboratoire d'Excellence (LABEX) CEBA (ANR-10-LABX-2501) and LABEX TULIP (ANR-10-LABX-0041), both managed by the Agence Nationale de la Recherche in France. PG received additional support from Fundação para a Ciência e a Tecnologia through the project BUSHRISK (IC&DT 02/SAICT/2017-032130).

## Author contributions

RFB: methodology, analysis, writing – original draft; LDB: methodology, analysis, writing – original draft; ABO: methodology, analysis, writing – review & editing; MSR: methodology & analysis; XL: methodology & analysis; GB: sampling, writing – review & editing; JS: writing – review & editing; CGS: analysis, writing – review & editing; SH: analysis; DZ: writing – review & editing; MP: sampling, writing – review & editing; VM: sampling; CM: sampling; MS: methodology, writing – review & editing; JK: analysis; LQ: analysis; GGE: analysis, writing – review & editing; FFS: analysis; RR: analysis; MH: analysis; LL: analysis; XW: analysis; MPH: sampling, writing – review & editing; TPLS: sampling, writing – review & editing; KH: analysis; MSS: sampling, writing – review & editing; A. Atickem: sampling; LC: writing – review & editing; CR: analysis, writing – review & editing; PG: sampling, writing – review & editing; HRS: sampling, writing – review & editing; IM: supervision, writing – original draft; A. Albrechtsen: supervision, writing – original draft, RH: supervision, writing – original draft. All authors proofread and approved the final version of the manuscript.

## Competing interests

The authors declare no competing interests.

## Additional information

Renzo F. Balboa [1,15], Laura D. Bertola [1,15], Anna Brüniche-Olsen [1,15], Malthe Sebro Rasmussen [1], Xiaodong Liu [1], Guillaume Besnard[2], Jordi Salmona[2], Cindy G. Santander [1], Shixu He [1], Dietmar Zinner [3,4,5], Miguel Pedrono[6], Vincent Muwanika [7], Charles Masembe [8], Mikkel Schubert [1,9], Josiah Kuja[1], Liam Quinn [1], Genís Garcia-Erill [1], Frederik Filip Stæger [1], Rianja Rakotoarivony[6], Margarida Henrique[10], Long Lin[1], Xi Wang[1], Michael P. Heaton [11], Timothy P. L. Smith [11], Kristian Hanghøj[1], Mikkel-Holger S. Sinding [1], Anagaw Atickem [12], Lounès Chikhi[2,10], Christian Roos [13], Philippe Gaubert[2,14], Hans R. Siegismund[1], Ida Moltke [1,16] ✉, Anders Albrechtsen [1,16] ✉ & Rasmus Heller [1,16] ✉

[1]Department of Biology, University of Copenhagen, Copenhagen, Denmark. [2]Laboratoire Evolution et Diversité Biologique (EDB), UMR 5174, CNRS, IRD, Université Toulouse Paul Sabatier, 31062 Toulouse, France. [3]Cognitive Ecology Laboratory, German Primate Center, Leibniz Institute for Primate Research, 37077 Göttingen, Germany. [4]Department of Primate Cognition, Georg-August-Universität Göttingen, 37077 Göttingen, Germany. [5]Leibniz Science Campus Primate Cognition, 37077 Göttingen, Germany. [6]UMR ASTRE, CIRAD, Campus International de Baillarguet, Montpellier, France. [7]College of Agricultural and

Environmental Sciences, Makerere University, Kampala, Uganda. [8]College of Natural Sciences, Makerere University, Kampala, Uganda. [9]Novo Nordisk Foundation Center for Basic Metabolic Research, University of Copenhagen, Copenhagen, Denmark. [10]Instituto Gulbenkian de Ciência, Oeiras, Portugal. [11]USDA, ARS, US Meat Animal Research Center, Clay Center, NE, USA. [12]Department of Zoological Sciences, Addis Ababa University, PO Box 1176 Addis Ababa, Ethiopia. [13]Gene Bank of Primates and Primate Genetics Laboratory, German Primate Center, Leibniz Institute for Primate Research, 37077 Göttingen, Germany. [14]Centro Interdisciplinar de Investigação Marinha e Ambiental (CIIMAR), Universidade do Porto, Terminal de Cruzeiros do Porto de Leixões, Av. General Norton de Matos, s/n, 4450-208 Porto, Portugal. [15]These authors contributed equally: Renzo F. Balboa, Laura D. Bertola, Anna Brüniche-Olsen. [16]These authors jointly supervised this work: Ida Moltke, Anders Albrechtsen and Rasmus Heller. ✉e-mail: ida@bio.ku.dk; aalbrechtsen@bio.ku.dk; rheller@bio.ku.dk

