## [Peer Review File · Nature Communications]

African bushpigs exhibit porous species boundaries and appeared in Madagascar concurrently with human arrivalReviewers' Comments:

Reviewer #1:

Remarks to the Author:

The manuscript on African bush pigs is a very detailed and thorough study of the two *Potamochoerus* species in Africa and based on the complete genome sequence of 67 individuals from across Africa, including Madagascar. The authors have used state of the art methodology to study the population genetics, speciation and admixture of these species. I have few comments on this manuscript which describes a very coherent and convincing story about these two species. In their discussion, the authors briefly touch upon the ongoing debate about the taxonomy of *Potamochoerus* and I agree with their comment that this very much depends on the species concept used. In fact, the species concept reflects the urge of humans to categorize, while the past decades have increasingly shown the diffuse borders between species with ongoing hybridization between many populations assumed to represent different species. I also welcomed their cautionary remark about the interpretation of PSMC curves and the cautionary remark about the assumption of total panmixia which indeed not always might be correct.

Minor comments:

Lines 201-203. I found the comment about the lowest D_{xy} between Ugandan and Congolese populations because of the closest geographical distance between these populations not very convincing. The distance between several others (e.g. Gabon) as well as the actual D_{xy} values between Ugandan and several others were not that very much different. Consider removing this sentence.

Figure 3c. The authors do not very clearly indicate why only these 18 individuals are shown. Although the authors mention it is based on read depth, they could have been a bit clearer and also might have used a read depth of $>10x$ as the threshold and shown the 23 individuals that are above that threshold.

Reviewer #2:

Remarks to the Author:

Balboa et al present a fascinating study on the evolutionary history of African suids. This manuscript specifically interrogates the putative suture zone of red river hogs and bushpigs, including estimating levels of admixture and divergence times. In addition, the manuscript focusses on the Malagasy bushpig divergence, and how this likely relates to historic human wildlife translocations.

The study is well-presented, and appealing to a wide audience, with broader relevance to phylogenetic patterns in Africa, factors influencing speciation and human-influenced species distributions.

I have questions around some of the analyses/interpretations that I would like the authors to clarify before I can recommend publication:

METHODS

The "Runs of homozygosity" section:

"at least two heterozygous reads to make a heterozygous call."

- What do you mean heterozygous reads? Do you mean allelic balance of ≥ 0.2 ? I.e. at least 2 alleles out of 10 being e.g. alternate?

"SNP sites with $> 50\%$ heterozygous genotypes across individuals were also excluded."

- Why? So you are filtering out regions of high heterozygosity? Presumably because you are assuming

that these correspond to repeat regions / duplications? Shouldn't this already be taken into account with your repeat masking that you have already done on the genome?

Have you considered a slightly more sophisticated program for identifying RoH's? E.g. RoHan? If you don't believe this is necessary/appropriate - why not?

TT and split time estimations

- This all sounds great, however, I have not used this method before and so it is difficult for me to judge the robustness of the analysis. The reference explains the theory of the approach, but some expansion of how the theory has been applied to this specific case would be very useful to the reader, for example in the supp methods? Or if not, at least a reference to a different paper that has used the approach and has given a more complete description.

RESULTS

"with only the Congo individuals being closer to the bushpigs than the other red river hogs"

- I initially read this as you saying that the Congo individuals were closer to bushpigs than red river hogs, but actually I think you mean "were closer to bushpigs than the other red river hogs, but overall still closer to red river hogs". Perhaps consider rewording it to clarify (as long as you can find a way that is better than my suggestion!)

In222-229 "Given the observed F_{ST} and D_{xy} values..."

- In this paragraph you appear to shift back and forth when talking about between "species" ranges to within. When you then end with Uganda it is not clear which of these two you are talking about

For the Dstatistic analysis, you state "Madagascar as H1" in the Figure 2c legend and "non-Malagasy bushpigs (H2)", but the figure shows "SA" as H1. Also in the text In242, you state "Madagascar as H1". Please clarify.

Could the "strength" of the signal you detect, not just be reflecting the power of the analysis for each pairwise comparison, rather than the amount of gene-flow itself? Could it be worth calculating f_4 ratios to attempt to quantify the admixture proportions for these pairwise comparisons?

"Given the results reported above, the unique demographic histories in Uganda and Ethiopia could be influenced by their geographic location as a place of introgression between the two taxa."

- This seems a bit speculative to me. The PSMC results are interesting, and visually appealing, and therefore I do think that they add to the story, however they are difficult to interpret in terms of any kind of specific hypothesis. Consider re-wording or removing part of this section.

"This analysis suggested that the Malagasy population experienced a severe bottleneck, likely a result of a founder event between 1-5 kya."

- The decline seems to start quite some time before this though - why might this be?

DISCUSSION

"had led to suggestions of multiple, distinct introduction pulses through the Comoros Islands and the North Mozambique current 63. However, from the PCA, NGAadm and IBS tree we did not identify substantial structure within the island, which is consistent with a relatively homogeneous founder population."

- To?: "...North Mozambique current 63. However, while we were unable to explicitly test this hypothesis with our dataset (due to no samples from Comoros etc), PCA NGAadm and the IBS tree did not.."?

Reviewer #3:

Remarks to the Author:

The paper is clearly written and illustrated, though as an archaeologist I am in no position to pass judgement on the detailed methodology of the genetic study undertaken. That said, references to the debate surrounding the timing of human settlement of Madagascar are fair and the conclusion that bushpigs were likely introduced there ~1000-1500 years ago does indeed fit extremely well with: a) dated faunal remains from the island, not just of bushpig but also of other exotic mammal taxa; b) the oldest archaeologically unambiguous, well-dated evidence for human presence; c) and specifically the African origins (and names) of domestic mammals and evidence for Triangular Incised Ware (Tana Ware), a form of pottery found in late first-millennium AD contexts in the southwest of the island - see Parker Pearson et al. 2010: 79 (Pastoralists, Warriors and Colonists: The Archaeology of Southern Madagascar; Oxford: Archaeopress).

For further discussion/reference of the timing of human settlement I would recommend Mitchell, *Journal of Island and Coastal Archaeology* 2020, arguments further elaborated in his 2022 book *African Islands: A Comparative Archaeology* (London: Routledge).

Provided that the genetic analysis is sound, I recommend acceptance.

Response to reviewer comments

A note to the Reviewers:

We thank Reviewers #1, #2 and #3 for taking time to review our manuscript and for the constructive and positive feedback that they have provided. We have detailed our point-by-point responses in red below and have provided a revised version of our manuscript. Line numbers in this response document refer to the line numbers in the original submitted manuscript.

REVIEWER COMMENTS

Reviewer #1 (Remarks to the Author):

The manuscript on African bush pigs is a very detailed and thorough study of the two *Potamochoerus* species in Africa and based on the complete genome sequence of 67 individuals from across Africa, including Madagascar. The authors have used state of the art methodology to study the population genetics, speciation and admixture of these species. I have few comments on this manuscript which describes a very coherent and convincing story about these two species. In their discussion, the authors briefly touch upon the ongoing debate about the taxonomy of *Potamochoerus* and I agree with their comment that this very much depends on the species concept used. In fact, the species concept reflects the urge of humans to categorize, while the past decades have increasingly shown the diffuse borders between species with ongoing hybridization between many populations assumed to represent different species. I also welcomed their cautionary remark about the interpretation of PSMC curves and the cautionary remark about the assumption of total panmixia which indeed not always might be correct.

Response: Thank you for taking the time to review our manuscript and for your positive comments.

Minor comments:

Lines 201-203. I found the comment about the lowest D_{xy} between Ugandan and Congolese populations because of the closest geographical distance between these populations not very convincing. The distance between several others (e.g. Gabon) as well as the actual D_{xy} values between Ugandan and several others were not that very much different. Consider removing this sentence.

Response: Thank you for your comment. We have updated lines 200-203 to be more precise in our description, where we include values and describe between-species D_{xy} relative to within-species D_{xy} between DR Congo and Ghana, as below:

“When comparing across species, D_{xy} was lowest for populations that were geographically central, with the Ugandan population exhibiting the lowest between-species D_{xy} for all bushpigs and the Congolese the lowest D_{xy} for all red river hogs. Additionally, the lowest D_{xy} between species was observed between the Ugandan and Congolese populations (0.00355), similar to within-species D_{xy} comparisons for Ghana and DR Congo (0.00352).”

Figure 3c. The authors do not very clearly indicate why only these 18 individuals are shown. Although the authors mention it is based on read depth, they could have been a bit clearer and also might have used a read depth of >10x as the threshold and shown the 23 individuals that are above that threshold.

Response: We agree that we were unclear in our description of medium-high depth samples. We used a minimum of 12X as our threshold since PSMC curves exhibit high variance when using samples $\leq 10X$ in combination with missing data (Nadachowska-Brzyska et al. (2016); ref. 71). As a result, we had 18 unrelated individuals that met this threshold, where the sample with the lowest coverage was 14X (RRivGab00178; Supplementary Data 1). Note that the same 18 individuals were consistently used for analyses which required medium-high depth and utilised genotype calls (D/f-branch statistics, PSMC, ROH, outgroup f3 and TT; described in Supplementary Data 1).

We have therefore added information in Methods ('Sequencing and mapping'; line 493) to reflect why these 18 individuals were chosen for specific analyses:

"As downstream analyses such as PSMC exhibit high variance when using lower-coverage samples ($\sim \leq 10\times$)⁷¹, we utilised a threshold of $>12\times$ depth of coverage to classify 18 unrelated individuals ($\geq 14\times$) as medium-high depth samples within this study (Supplementary Data 1)."

In addition, we have added information to the figure legends in all main figures so that they include "*medium-high depth ($\geq 14\times$)*".

Reviewer #2 (Remarks to the Author):

Balboa et al present a fascinating study on the evolutionary history of African suids. This manuscript specifically interrogates the putative suture zone of red river hogs and bushpigs, including estimating levels of admixture and divergence times. In addition, the manuscript focusses on the Malagasy bushpig divergence, and how this likely relates to historic human wildlife translocations.

The study is well-presented, and appealing to a wide audience, with broader relevance to phylogenetic patterns in Africa, factors influencing speciation and human-influenced species distributions.

I have questions around some of the analyses/interpretations that I would like the authors to clarify before I can recommend publication:

Response: Thank you for taking the time to review our manuscript and for the positive and constructive comments.

METHODS

The "Runs of homozygosity" section:

"at least two heterozygous reads to make a heterozygous call."

- What do you mean heterozygous reads? Do you mean allelic balance of ≥ 0.2 ? I.e. at least 2 alleles out of 10 being e.g. alternate?

Response: Yes, this was an error on our part; thank you for catching this. We meant at least two reads carrying each of the two alleles at heterozygous sites. We have now updated the main text in lines 644-647 to:

*"Runs of homozygosity (ROH) analyses were performed using PLINK v1.9⁸⁶. PLINK files included only filtered variable sites within medium-high depth samples ($n = 18$), with an additional depth filter (10 reads minimum) **and at least two reads carrying each of the two alleles at heterozygous sites**".*

"SNP sites with $> 50\%$ heterozygous genotypes across individuals were also excluded."

- Why? So you are filtering out regions of high heterozygosity? Presumably because you are assuming that these correspond to repeat regions / duplications? Shouldn't this already be taken into account with your repeat masking that you have already done on the genome?

Response: Before performing downstream analyses, we removed sites where almost all individuals in our entire dataset were expected to be heterozygous ($F < -0.9$) as such sites were very likely error prone. As the ROH analysis is particularly sensitive to spurious heterozygous sites, where even a single heterozygous site will break up a ROH, this specific analysis uses a smaller subset of individuals (the 18 medium-high depth individuals) and a more stringent filter for the retention of sites with many heterozygous genotypes.

This is now reflected in the main text, where lines 650-651 have been updated to:

"A few SNP sites with $> 50\%$ heterozygous genotypes across individuals were also excluded since these were prone to genotype errors which can break up longer ROHs."

Have you considered a slightly more sophisticated program for identifying RoH's? E.g. ROHan? If you don't believe this is necessary/appropriate - why not?

Response: ROHan is a great software created by one of the co-authors on this paper, so we are glad that you have suggested it :-). That being said, ROHan will most likely not give better results than PLINK for the high-depth individuals. However, it does allow us to infer ROHs for low-depth samples. So we agree that it is a useful analysis, providing a more complete picture of ROHs in the studied populations.

We have now updated the manuscript to include the analysis of all 67 individuals using ROHan, and have presented the results as part of Supplementary Figure S8. We observed similar trends to our previous ROH analyses. However, we caution that due to its window-based approach, ROHan has a hard time identifying shorter ROHs ($< 2\text{Mb}$) and will underestimate the length of most ROHs. So compared to PLINK-based ROH analyses, the overall length of ROHs from ROHan is slightly shorter but the length of the longer ROHs

(>2Mb) is very similar. We have added illustrations of the ROHs inferred by ROHan on the Github repository for this study.

We have updated lines 290-292 to include this information:

“Additionally, we explored ROH in all 67 samples, including low-depth samples with ROHan⁴³, which overall yielded similar results. However, due to its window-based approach, ROHan could not identify most of the shorter ROHs (<2 Mb), and therefore the overall F_{ROH} was slightly smaller using this approach (Supplementary Fig. S8). When considering both analyses, we observed heterogeneous levels of ROH within Uganda, Ethiopia, Cameroon and Equatorial Guinea likely driven by recent inbreeding events, leading to differences in longer ROHs.”

TT and split time estimations

- This all sounds great, however, I have not used this method before and so it is difficult for me to judge the robustness of the analysis. The reference explains the theory of the approach, but some expansion of how the theory has been applied to this specific case would be very useful to the reader, for example in the supp methods? Or if not, at least a reference to a different paper that has used the approach and has given a more complete description.

Response: Thank you for reminding us that not all of the methods we used are widely known! TT and very similar methods estimating split times have been used in multiple studies, with the first variant of the method being introduced in Rasmussen et al. (2014) (ref. 92). The method was named TT in Schlebusch et al. (2017) (ref. 46), where it was used to estimate the divergence times for human populations, and the method paper we reference, Sjödin et al. (2021) (ref. 45), then came afterwards to describe and test the method in more detail. Subsequent studies have then further used this approach, including Larena et al. (2021) (ref. 89), Hollfelder et al. (2021) (ref. 90), Garcia-Erill et al. (2022) (ref. 4), Oliveira et al. (2023) (ref. 91) and Quinn et al. (2023) (ref. 93).

To make the interpretation easier for the reader, we have made a “tree” inside Figure 4b and have updated the figure legend to clarify what T1 and T2 describe. We have also updated the Methods section with additional detail (from line 667):

“Population split times were estimated using the Two-Two (TT) method^{45,46}. This approach estimates separate split times from a common ancestor for two different populations and has been used in various studies, including human^{46,89–91}, direct ancestry⁹² and animal studies^{4,93}. T1 describes the estimated time to the common ancestor for population 1 and T2 the time to the common ancestor for population 2. Further details for the method are described in Sjödin et al.⁴⁵ and its relationship with other similar methods in Mualim et al.⁹⁴. For this analysis, we utilised the unfolded 2D-SFS from medium-high depth individuals polarised using several outgroups: against the common warthog, desert warthog (Supplementary Data 1) and the domestic pig (SRA: SAMN28197093). A mutation rate of $1.49e^{-8}$ per site per generation and a generation time of six years were used to scale split times, as in our PSMC analyses^{4,42}.”

RESULTS

"with only the Congo individuals being closer to the bushpigs than the other red river hogs"

- I initially read this as you saying that the Congo individuals were closer to bushpigs than red river hogs, but actually I think you mean "were closer to bushpigs than the other red river hogs, but overall still closer to red river hogs". Perhaps consider rewording it to clarify (as long as you can find a way that is better than my suggestion!)

Response: Yes, this is what we meant - that these individuals are closer to the bushpigs when compared to the other red river hogs. We have now reworded lines 165-169 to the following (and hope that it is clearer!):

*"Principal component analysis (PCA) revealed that the first two principal components exhibited a spatial distribution pattern reflecting the taxonomic and geographic origins of the sampled pigs (Fig. 1b). **All the red river hog samples clustered together, yet the Congolese individuals were closer to the bushpigs than the other red river hogs.** Malagasy samples formed a separate cluster from the other bushpigs."*

In222-229 "Given the observed F_{ST} and D_{xy} values..."

- In this paragraph you appear to shift back and forth when talking about between "species" ranges to within. When you then end with Uganda it is not clear which of these two you are talking about

Response: We have now clarified this by rewriting lines 222-229 as:

"Given the observed F_{ST} and D_{xy} values, we explored spatial population structure and gene flow between and within the two species through estimating effective migration rates (Fig. 2b)³⁸. Between-species comparisons revealed a general barrier through the Central African rainforest and the East African Rift Valley, separating W/C and E/S populations. Within-species comparisons revealed high connectivity within both bushpig and red river hog ranges respectively, with the exception of Malagasy and non-Malagasy bushpigs which exhibited a barrier across the Mozambique Channel, particularly with the northernmost non-Malagasy populations. A decrease in effective migration in Ethiopia was also observed; this is in contrast with Uganda where we observed weak barriers, suggesting a possible corridor of gene flow connectivity within this region."

For the Dstatistic analysis, you state "Madagascar as H1" in the Figure 2c legend and "non-Malagasy bushpigs (H2)", but the figure shows "SA" as H1. Also in the text In242, you state "Madagascar as H1". Please clarify.

Response: Thank you for catching this - it was indeed an error on our part with respect to the main text and the Fig. 2c legend. We have corrected the main text (line 242; see below) and Fig. 2c (removed 'non-Malagasy') so that these only describe South Africa as H1.

*"Similarly, to test for gene flow in the opposite direction, we performed similar tests with the **southernmost** bushpig population, **South Africa** as H1, each of the remaining red river hog populations as H2, and each of the bushpig populations as H3 (Fig. 2c; lower panel)."*

Could the "strength" of the signal you detect, not just be reflecting the power of the analysis for each pairwise comparison, rather than the amount of gene-flow itself? Could it be worth

calculating f_4 ratios to attempt to quantify the admixture proportions for these pairwise comparisons?

Response: We agree that the D-statistic value by itself is hard to interpret and does not translate well to determining the amount of gene flow. As suggested, we have calculated f_4 -ratios to quantify the amount of gene flow within Uganda and other populations (described below); these results show similar patterns to D-statistics analyses.

Figure above (now Supplementary Fig. S7): Estimation of admixture proportions (f_4 -ratios) between populations.

f_4 -ratios were calculated using the common warthog as an outgroup, constructed as $f_4(A,O;X,C)/f_4(A,O;B,C)$. **a)** f_4 -ratios into Uganda from non-Ghanian red river hog populations (B), of the form $(f_4(\text{Ghana}, \text{Warthog}; \text{Uganda}, \text{South Africa})/f_4(\text{Ghana}, \text{Warthog}; \text{B}, \text{South Africa}))$. **b)** f_4 -ratios into Ethiopia from non-Ghanian red river hog populations (B) $(f_4(\text{Ghana}, \text{Warthog}; \text{Ethiopia}, \text{South Africa})/f_4(\text{Ghana}, \text{Warthog}; \text{B}, \text{South Africa}))$. **c)** f_4 -ratios into bushpig

populations (X), using Equatorial Guinea as a proxy. α indicates estimated admixture proportions from B' into X ($f_4(\text{Ghana, Warthog; X, South Africa})/f_4(\text{Ghana, Warthog; Eq Guinea, South Africa})$). Error bars represent \pm three standard errors from the estimated f_4 -ratio. SA - South Africa; EqG - Equatorial Guinea.

In the figure above, we describe three scenarios: a) where we estimate admixture proportions into **Uganda** from non-Ghanian red river hog populations, b) where we estimate admixture proportions into **Ethiopia** from non-Ghanian red river hog populations, and c) where we estimate admixture proportions into each of the bushpig populations from Equatorial Guinea red river hogs.

As can be seen in the first scenario (a), we show that there is an increase in f_4 -ratios for the Uganda population as one moves from the westernmost part of Africa (Togo; 10.9%) to the most central (Gabon; 21.1%), in line with our hypothesis that Uganda is a potential zone of introgression and in agreement with the large signals observed in D-statistics comparisons (Fig. 2c). This is similarly seen for Ethiopia (b), with estimated admixture proportions of 6.8% to 13.2%, from west (Togo) to central (Gabon), respectively.

In the third scenario (c), we observe high levels of gene flow into Uganda and Ethiopia (19.7% and 12.2%, respectively) and much lower (albeit significant) levels in Tanzania and Zimbabwe, supporting our results in the manuscript.

Overall, these results, combined with the D-statistics, suggest significant gene flow into Uganda and Ethiopia from red river hog populations or one or more populations closely related to them that were not included in this study. We have added the plot to Supplementary Information as Supplementary Fig. S7.

Although the analysis infers ~20% gene flow, it is worth noting that there are assumptions that the populations considered had evolved as a perfectly bifurcating tree and that the only gene flow event is the one that is modelled. As suggested by our f -branch (f_b) results (Supplementary Fig. S6), the relationship between the populations is likely much more complicated, and therefore one should be cautious in interpreting these results as evidence for a single point migration event. The amount of gene flow could be much higher from an unsampled population or it could be many migration events happening between several populations at different time points.

In light of these results, we have added the following to the main text (line 247):

“Furthermore, we estimated the amount of gene flow between species based on f_4 -ratios, under assumptions that the populations considered had evolved together as a perfectly bifurcating tree and that the only gene flow event is the one that is modelled. This analysis estimated up to 21% gene flow from red river hogs into Uganda and up to 13% into Ethiopia, with increasing signals in more central populations (Supplementary Fig. S7). Given the complicated history of these populations, as suggested by our f -branch (f_b) results (Supplementary Fig. S6), these values are unlikely to accurately represent the historical gene flow events which likely occurred between multiple populations at different timepoints. Nevertheless, when considering all three analyses, these results suggest that there is or has been gene flow between the two taxa currently identified as species, and that the gradient of

allele sharing between them is consistent with isolation by distance, where genetic similarity is strongest in populations from Central Africa.”

"Given the results reported above, the unique demographic histories in Uganda and Ethiopia could be influenced by their geographic location as a place of introgression between the two taxa."

- This seems a bit speculative to me. The PSMC results are interesting, and visually appealing, and therefore I do think that they add to the story, however they are difficult to interpret in terms of any kind of specific hypothesis. Consider re-wording or removing part of this section.

Response: The above statement was not meant to be purely based on PSMC; rather, this was meant to be in light of the D-statistics (and now f_4 -ratio) results that were described before this section. We have therefore revised the text to reflect this (lines 263-264):

“These results, in combination with the D-statistics and f_4 -ratio results reported above, suggest that the unique demographic histories in Uganda and Ethiopia could be influenced by their geographic location as a place of possible introgression between the two taxa.”

"This analysis suggested that the Malagasy population experienced a severe bottleneck, likely a result of a founder event between 1-5 kya."

- The decline seems to start quite some time before this though - why might this be?

Response: The decline in effective population size does appear to start in the interval ~5-8kya where it drops to ~20,000 individuals, followed by the large decline to ~1,000 individuals around 1kya. When replying to this comment, we realised how difficult it was to see when the decline happened due to the log scale, so we have subsequently updated the figure with additional tick marks. It is hard to say whether the early start of the decline is due to a decline in the population in mainland Africa prior to colonisation of Madagascar or whether it is a methodological issue; this is now reflected in the main text.

We have rewritten the relevant part of the Discussion (lines 418-429) as:

“Our popSizeABC estimate of an effective population size of 1,000 individuals during the bottleneck 1,500 years ago is surprisingly high, assuming that the founder event was a single occurrence involving a limited number of individuals carried to Madagascar by ship. However, the estimate is supported by their observed heterozygosity level, which is similar to levels observed in southern Africa, although we cannot know to what extent southern African populations have been subjected to drift since the founding of the Malagasy population. Multiple introductions, spanning over a longer period of time, or even with animals sourced from different mainland populations (including already admixed individuals), may have inflated this estimate. In addition, an instant large drop in population size followed by subsequent regrowth will be estimated by popSizeABC as a more gradual decline that starts earlier⁴⁶, making it difficult to pinpoint the exact timing of when the bottleneck started.”

DISCUSSION

"had led to suggestions of multiple, distinct introduction pulses through the Comoros Islands and the North Mozambique current 63. However, from the PCA, NGAadmix and IBS tree we

did not identify substantial structure within the island, which is consistent with a relatively homogeneous founder population."

- To?: "...North Mozambique current 63. However, while we were unable to explicitly test this hypothesis with our dataset (due to no samples from Comoros etc), PCA NGAdmix and the IBS tree did not.." ?

Response: We agree that we should have made it clear that we could not have made statements about bushpig populations in the Comoros Island, since we do not have data from that area. We have therefore incorporated your suggestion and have changed lines 446-449 to:

*"...had led to suggestions of multiple, distinct introduction pulses through the Comoros Islands and the North Mozambique current ⁶⁷. **Although we were unable to explicitly test this hypothesis without samples from the Comoros, our results from the PCA, NGSadmix and IBS tree do not suggest multiple pulses into Madagascar and did not identify substantial structure within the island, which is consistent with a relatively homogeneous founder population.**"*

Reviewer #3 (Remarks to the Author):

The paper is clearly written and illustrated, though as an archaeologist I am in no position to pass judgement on the detailed methodology of the genetic study undertaken. That said, references to the debate surrounding the timing of human settlement of Madagascar are fair and the conclusion that bushpigs were likely introduced there ~1000-1500 years ago does indeed fit extremely well with: a) dated faunal remains from the island, not just of bushpig but also of other exotic mammal taxa; b) the oldest archaeologically unambiguous, well-dated evidence for human presence; c) and specifically the African origins (and names) of domestic mammals and evidence for Triangular Incised Ware (Tana Ware), a form of pottery found in late first-millennium AD contexts in the southwest of the island - see Parker Pearson et al. 2010: 79 (Pastoralists, Warriors and Colonists: The Archaeology of Southern Madagascar; Oxford: Archaeopress).

For further discussion/reference of the timing of human settlement I would recommend Mitchell, Journal of Island and Coastal Archaeology 2020, arguments further elaborated in his 2022 book African Islands: A Comparative Archaeology (London: Routledge).

Provided that the genetic analysis is sound, I recommend acceptance.

Response: Thank you for spending time reviewing our manuscript and for providing us with important references to include. To further guide the discussion about human arrival, we have now incorporated both the Mitchell (2020) and Parker Pearson (2010) references in our Discussion, in line 434:

"...most likely through populations that started to arrive on Madagascar from southeastern Africa at least 1,500 years ago and possibly earlier ²⁶, although the latter has been contested ⁶⁴. Triangular Incised Ware (Tana Ware) style pottery remains were found in human settlements in southern Madagascar dated to 1,000-1,400 years ago, suggesting either contact with or colonisation from the African Swahili region at this time ⁶⁵."

Reviewers' Comments:

Reviewer #1:

Remarks to the Author:

I already only had small number of minor comments on the original manuscript, and these have all been sufficiently addressed in the revised manuscript.

Reviewer #2:

Remarks to the Author:

All my comments/questions have been addressed. The authors should congratulate themselves on producing an well-written and important piece of work!

REVIEWERS' COMMENTS

Reviewer #1 (Remarks to the Author):

I already only had small number of minor comments on the original manuscript, and these have all been sufficiently addressed in the revised manuscript.

Response: We are pleased to hear that – thank you for your time and effort in reviewing our manuscript!

Reviewer #2 (Remarks to the Author):

All my comments/questions have been addressed. The authors should congratulate themselves on producing an well-written and important piece of work!

Response: We appreciate the kind comment and for taking the time to provide positive and constructive comments to help improve our manuscript!